# FlingBot: The Unreasonable Effectiveness of Dynamic Manipulation for Cloth Unfolding

**Huy Ha**     **Shuran Song**
Columbia University
https://flingbot.cs.columbia.edu

**Abstract:** High-velocity dynamic actions (e.g., fling or throw) play a crucial role in our everyday interaction with deformable objects by improving our efficiency and effectively expanding our physical reach range. Yet, most prior works have tackled cloth manipulation using exclusively single-arm quasi-static actions, which requires a large number of interactions for challenging initial cloth configurations and strictly limits the maximum cloth size by the robot's reach range. In this work, we demonstrate the effectiveness of dynamic flinging actions for cloth unfolding with our proposed self-supervised learning framework, FlingBot. Our approach learns how to unfold a piece of fabric from arbitrary initial configurations using a pick, stretch, and fling primitive for a dual-arm setup from visual observations. The final system achieves over 80% coverage within 3 actions on novel cloths, can unfold cloths larger than the system's reach range, and generalizes to T-shirts despite being trained on only rectangular cloths. We also finetuned Fling-Bot on a real-world dual-arm robot platform, where it increased the cloth coverage over 4 times more than the quasi-static baseline did. The simplicity of FlingBot combined with its superior performance over quasi-static baselines demonstrates the effectiveness of dynamic actions for deformable object manipulation.

**Keywords:** Dynamic manipulation, Cloth manipulation, Self-supervised learning

## 1 Introduction

High-velocity dynamic actions play a crucial role in our everyday interaction with deformable objects. Making our beds in the morning is not effectively accomplished by picking up each corner of the blanket and placing them in the corresponding corners of the bed, one by one. Instead, we are more likely to grasp the blanket with two hands, stretch it, and then unfurl it with a fling over the bed. This fluid high-velocity flinging action is an example of dynamic manipulation [1], which is used to improve our physical reachability and action efficiency – allowing us to unfold large crumpled cloths with as little as one interaction.

From goal-conditioned folding [2] to fabric smoothing [3, 4], prior works have achieved success using exclusively single-arm quasi-static interactions (e.g., pick & place) for cloth

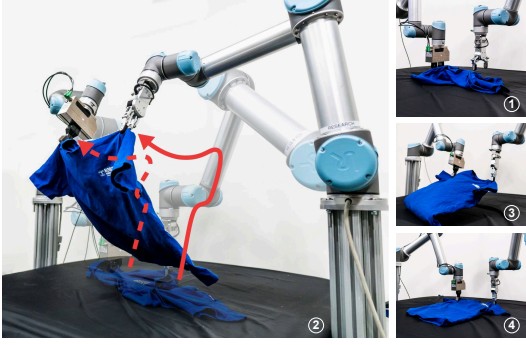

Figure 1: **Cloth unfolding with dynamic interactions.** Given a severely crumpled cloth, Fling-Bot uses a high-speed fling to unfurl the cloth with as little as one interaction. In this paper, we demonstrate that such dynamic actions can efficiently unfold cloths, generalizable to different cloth types, and improve the effective reach range of the system.

manipulation. However, these approaches require a large number of interactions for challenging initial configurations (e.g., highly crumpled cloths) or rely on strong assumptions about the cloth (e.g., predefined keypoints). Additionally, since the robot arm cannot manipulate the cloth at locations it can't reach, the maximum cloth size is greatly limited by the robot arm's reach range.

5th Conference on Robot Learning (CoRL 2021), London, UK.

In this work, we focus on the task of cloth unfolding, where the goal is to maximize the cloth's coverage. Because unfolding reveals key features of the cloth for downstream perception and manipulation, it is a typical first step for many cloth manipulation pipelines. An ideal cloth unfolding approach should be:

- **Efficient**: the approach should reach a high coverage with a small number of actions from arbitrarily crumpled initial configurations.

- **Generalizable**: the algorithm should not rely on heuristics (e.g., grasp predefined key points). This is especially important when unfolding is only the initial stage of a cloth perception and manipulation pipeline, where key points are not visible or severely occluded, and when the system must handle cloth types unseen during training, which may not contain the predefined key points.

- **Flexible Beyond the Workspace**: the approach should work with cloths of different sizes, including large ones which lie outside the robot's physical reach range.

To achieve this goal, we present FlingBot, a self-supervised algorithm that learns how to unfold cloths from arbitrary initial configurations using a pick, stretch, and fling primitive for a dual-arm setup. At each time step, the policy predicts value maps from its visual observation and picks actions greedily with respect to its value maps. To provide the supervision signal, the system computes the difference in coverage of the cloth before and after each action – the delta-coverage – from the visual input captured by a top-down camera. FlingBot achieves over 80% coverage within 3 actions on novel cloths and increases the cloth's coverage by more than twice that of pick & place and pick & drag quasi-static baselines on rectangular cloths. Our approach is flexible to large cloths whose dimensions exceed the robot arm's reach ranges and generalizes to T-shirts despite being trained on rectangular cloths. We fine-tune our approach in the real world, where, on average, it increased the cloth coverage over 4 times more than the quasi-static pick & place baseline did. In summary:

- Our main contribution is in demonstrating the effectiveness of dynamic manipulation for cloth unfolding through our self-supervised learning framework, FlingBot.

- We propose a parameterization for the dual-arm grasp of our fling primitive, which enables the application of a simple yet effective single-arm grasping technique [5, 6, 7] to dual-arm grasping while satisfying dual-arm safety constraints. The simplicity of FlingBot combined with its superior performance over quasi-static baselines further emphasize the effectiveness of dynamic actions for deformable object manipulation.

- We present a custom simulator[1] built on top of PyFlex [8, 9], a CUDA accelerated simulator, which supports the loading of arbitrarily shaped cloth meshes. We hope this open-source simulator expands cloth manipulation research to more complex cloth types.

## 2   Related Work

A convincing argument for the *addition* of dynamic actions to exclusively quasi-static cloth manipulation pipelines would need to demonstrate their superior performance on a core cloth manipulation skill. As a typical first step for many cloth manipulation tasks, cloth unfolding is a popular and important problem setting for studying deformable object manipulation. The goal of cloth unfolding is to maximize the coverage of the cloth on the workspace, which exposes key visual features of the cloth for downstream applications. However, achieving a fully unfolded cloth configuration from a crumpled initial configuration remains a challenging problem. Additionally, doing so efficiently for many different types of cloths, some larger than the system's reach range, is extremely challenging.

**Quasi-static Cloth Manipulation with Expert Demonstrations.** Prior works have explored using heuristics and identifying key points such as wrinkles [10], corners [11, 12, 13], edges [14], or combinations of them [15], but fail to generalize to severely self-occluded cloth configurations, to when required key points aren't visible, or to non-square cloths. Recent reinforcement learning approaches [3, 16] relied on cloth unfolding expert demonstrations in quasi-static pick-and-drag action

---

[1]Please visit https://flingbot.cs.columbia.edu for experiment videos, code, simulation environment, and data.

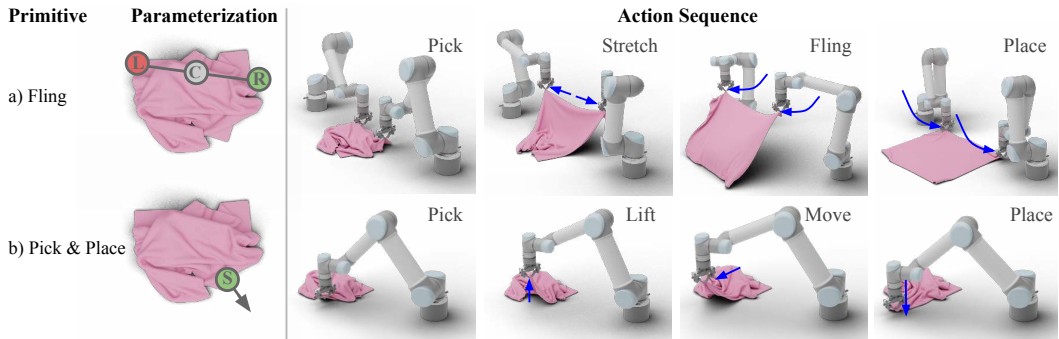

Figure 2: **Action Primitives.** The dynamic **Fling** primitive starts with a two-arm grasp at the left $L$ and right $R$ grasp locations with center point $C$, followed by a fixed stretch, fling, and place routine. A good fling could unfold a highly crumpled cloth in as little as a single step. In contrast, the quasi-static **Pick & Place** primitive, which grasps at the start location $S$, lifts, moves, then places at the end position specified by the arrow tip, requires many steps for such challenging cloth configurations.

spaces, While they sidestep the exploration problem, expert demonstrations can be sub-optimal and brittle if from hard-coded heuristics [3] or expensive if from humans [16].

**Self-supervised Quasi-static Cloth Manipulation.** Bypassing the dependency on an expert, self-supervised cloth manipulation has been demonstrated in unfolding with a factorized pick & place action space by Wu et al. [4] and goal conditioned folding using spatial action maps [7, 6, 5] by Lee et al. [2]. However, all these approaches operate entirely in quasi-static action spaces.

**Dynamic Cloth Manipulation.** In contrast to quasi-static manipulation, which are "operations that can be analyzed using kinematics, static, and quasi-static forces (such as frictional forces at sliding contacts)" [1], dynamic manipulation involves operations whose analysis additionally requires "forces of acceleration". Intuitively, dynamic manipulation (e.g., tossing [6]) involve high-velocity actions which build up objects' momentum, such that the manipulated objects continue to move after the robot's end-effector stops. Such dynamic manipulation results in an effective increase in reach range and a reduction in the number of actions to complete the task compared to exclusively quasi-static manipulation. However, prior works on dynamic cloth manipulation have either relied on cloth's vertex positions [17] (so is only practical in simulation), a motion capturing system with markers combined with human demonstrations [18], or custom hardware [19, 20].

In contrast, our algorithm achieves high performance from severely crumpled initial cloth configurations through self-supervised trial and error and learns directly from visual input without requiring expert demonstrations or ground truth state information.

## 3 Method

The goal of cloth unfolding is to manipulate a cloth from an arbitrarily crumpled initial state to a flattened state. Concretely, this amounts to maximizing the cloth's coverage on the workspace surface. Intuitively, dynamic actions have the potential to achieve high performance on cloth unfolding by appropriately making use of the cloth's mass in a high-velocity action to unfold the cloth (Sec. 3.1). From a top-down RGB image of the workspace with the cloth, our policy decides the next fling action (Sec. 3.2) by picking the highest value action which satisfies the system's constraints (Sec. 3.3). It predicts the value of each action with a value network (Sec. 3.4) which is trained in a self-supervised manner to take actions that maximally increases the cloth's coverage. The supervision signal is computed directly from the visual observation captured by the top-down camera. After training in simulation, we finetune and evaluate the model in the real world (Sec. 3.5).

### 3.1 Advantages of the Fling Action Primitive

Quasi-static actions, such as pick & place, rely on friction between the cloth and ground to stretch the cloth out. The complexity of friction forces between the workspace and the (nonvisible, ground-facing) surfaces of the cloth means the cloth's final configuration may be difficult to predict from

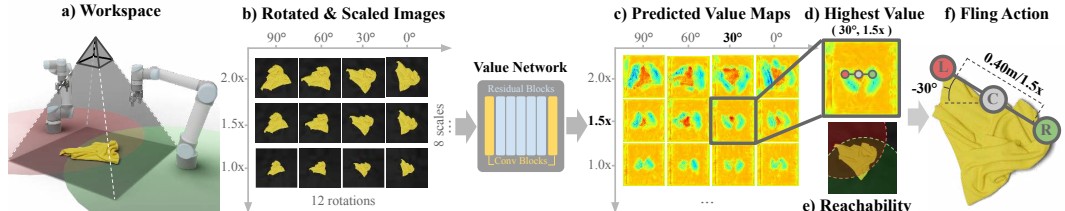

Figure 3: **Method Overview.** From a top-down RGB image a), our policy evaluates a batch of different action rotations and scales by transforming the observation b) then predicting the corresponding batch of value maps c). The highest value action d), corresponding to the maximum value pixel which also satisfies the arms' reachability constraints e) is chosen. Finally, the chosen pixel's location and its observation's transformation is decoded into the fling action parameters (i.e., center point, distance, relative orientation between gripper).

only visual observations, especially for novel cloth types or significantly different friction forces (i.e.: simulation v.s. real). In addition, such systems can't manipulate points on the cloth to regions outside of their physical reach range, which limits their maximum cloth dimensions. Meanwhile, dynamic actions largely rely on cloths' mass combined with a high-velocity throw to do most of its work. Since a single dynamic motion primitive can effectively unfold many cloths, dynamic unfolding systems can learn a simpler policy which generalizes better to different cloth types. In addition, they can reach higher coverages in smaller numbers of interactions and throw corners of cloths larger than the system's reach range, effectively expanding their physical reach range.

We propose to use a dual-arm pick, stretch, and fling primitive. Here, a dual-arm system stretches the cloth between the arms, which unfolds it in one direction, and then flings the cloth, which makes use of the cloth's mass to unfold it in the other direction. The combination of stretching and flinging ensures that, given two appropriate grasp points, our motion primitive should be sufficient for single step unfolding when possible.

### 3.2 Fling Action Primitive Definition

To achieve an efficient, generalizable, and flexible cloth unfolding system, we argue that the system requires two arms operating in a dynamic action space. To this end, we've designed a pick, stretch, and fling motion primitive for a dual-arm system, where each arm is placed on either side of the cloth workspace, as follows. First, the arms perform a top-down pinch-grasp on the cloth at locations $L, R \in \mathbb{R}^3$. Second, the arms lift the cloth to $0.30$m and stretch the cloth taut in between them. Third, the arms fling the cloth forward $0.70$m at $1.4 \mathrm{m\,s}^{-1}$ then pull backwards $0.50$m at $1.4 \mathrm{m\,s}^{-1}$, which swings the cloth forwards. Finally, the arms place the cloth down to the workspace and release their grips. The entire motion primitive is visualized in (Fig. 2a).

Instead of learning all steps of our primitive, we can fix the stretching step to always stretch the cloth as much as possible without tearing the cloth. We also fix the fling speed and trajectory from the observation that the real world system could robustly unfold the cloth using a wide range of fling parameters (i.e.: fling height, fling speed) if given a good grasp (see supplementary materials). Thus, the problem of learning to pick, stretch, and fling reduces down to the problem of learning where to pick, which is parameterized with only two grasp locations, one for each arm.

### 3.3 Constraint Satisfying Fling Action Parameterization

A dual-arm grasp is parameterized by two points $L, R \in \mathbb{R}^3$, which denotes where the left and right arm should approach from a top-down grasp respectively, and requires 6 scalars. Without loss of generality for the purposes of grasping from a top-down RGB-D input, the third dimension could be specified by depth information. This reduces $L$ and $R$ to two points in $\mathbb{R}^2$, each representing pixels to grasp from the visual input and uses only 4 scalars in total.

However, to minimize collisions between two arms, we wish to impose a constraint that $L$ is always left of $R$, and vice versa . Additionally, the grasp width (i.e., the length of the line $L - R$) be greater than the physical limit of the system and smaller than the minimum safe distance limit between the two arms. Directly using $L$ and $R$ will entangle these two constraints (see supplementary materials),

making them difficult to satisfy. To make these constraints linear and independent, we propose a 4-scalar parameterization, which consists of the pixel $C \in \mathbb{R}^2$ at the center of the line $L - R$, an angle $\theta \in \mathbb{R}$ denoting the planar rotation of the line $L - R$, and a grasp width $w \in \mathbb{R}$ denoting the length of the line in pixel space. To constrain $L$ to be on the left of $R$, we can constrain $\theta \in [-90°, 90°]$, while $w$ can be directly constrained to appropriate system limits.

## 3.4 Learning Delta-Coverage Maximizing Fling Actions

The naive approach for learning the action parameters $\langle C_x, C_y, \theta, w \rangle$ by directly predicting these 4 scalars does not accommodate the best grasp's equivariances to the cloth's translation, rotation, and scale. However, the learner should be equipped with the inductive bias that cloths in identical configurations have identical optimal grasp points for flinging relative to the cloth.

To this end, we propose to use spatial action maps [5, 6, 7]. By predicting grasp values with constant scale and rotation *in pixel space* from the transformed images, spatial action maps can recover grasp values with varying scales and rotations *in world space* by varying the transformation applied to the image. Concretely, given a visual observation from the top-down view (Fig. 3a), we generate a batch of rotated and scaled observations (Fig. 3b) then predict the corresponding batch of dense value maps (Fig. 3c). Each pixel in each value map contains the value of the action parameterized by that pixel's location, giving $C$, and its observation's rotation and scaling, giving $\theta$ and $w$ respectively (Fig. 3f). The value network is supervised to regress each pixel in the value map to the ground truth delta-coverage – the difference in coverage before and after the action. Thus, by picking the grasping action with the highest value (Fig. 3d), the system picks grasp points which it expects would lead to the greatest increase in cloth coverage. By its architecture, the value network is equivariant to translation. By rotating and scaling its inputs, it is also equivariant to rotation and scale.

We use 12 rotations in $[-90°, 90°]$, 8 scale factors in $[1.00, 2.75]$, both at even intervals, and $64 \times 64$ RGB images. To make the use of the scale range, we crop the image such that the cloth takes up two-thirds of the image width and height before applying one of the 8 scale factors. We also filter out and reject all grasp point pairs which are out of reach for either arm.

**Self-Supervised Learning.** The value network is trained end-to-end with self-supervised trials in simulation then finetuned in the real world. Each fling step is labeled with its normalized delta coverage, which is computed by counting cloth mask pixels from a top-down view, then dividing by the cloth mask pixel count of a flattened cloth state. While our experiments have plain black workspace surface colors and non-textured colored cloths for easy masking, our approach can be combined with image segmentation or background subtraction approach to obtain the corresponding cloth mask to work with arbitrary cloth and workspace textures. Note that this supervision signal (i.e., cloth mask) is only needed during training.

The policy interacts with the cloth until it reaches 10 timesteps or predicts grasps on the workspace. The latter stopping condition indicates that the policy does not expect any possible action to further improve the cloth coverage. All policies are trained in simulation until convergence, which takes around 150,000 simulation steps, or 6 days on a machine with a GTX 1080 Ti. The network is trained using the Adam optimizer with a learning rate of 1e-3 and a weight decay of 1e-6.

## 3.5 Experiment Setup

**Simulation Setup.** Our custom simulator[2] is built on top of the PyFleX [8] bindings to Nvidia FleX provided by SoftGym [9], and can load arbitrary cloth meshes, such as T-shirts, through a Python API. The observations are rendered using Blender 3D where the cloth HSV color is sampled uniformly between $[0.0, 1.0]$, $[0.0, 1.0]$, and $[0.5, 1.0]$ respectively, and the background is dark grey with a procedurally generated normal map to mimic the wrinkles on the real-world workspace surface. We further apply brightness, contrast, and hue jittering on observations to help with the transfer

---

[2] Our custom simulator is publicly accessible at https://github.com/columbia-ai-robotics/flingbot

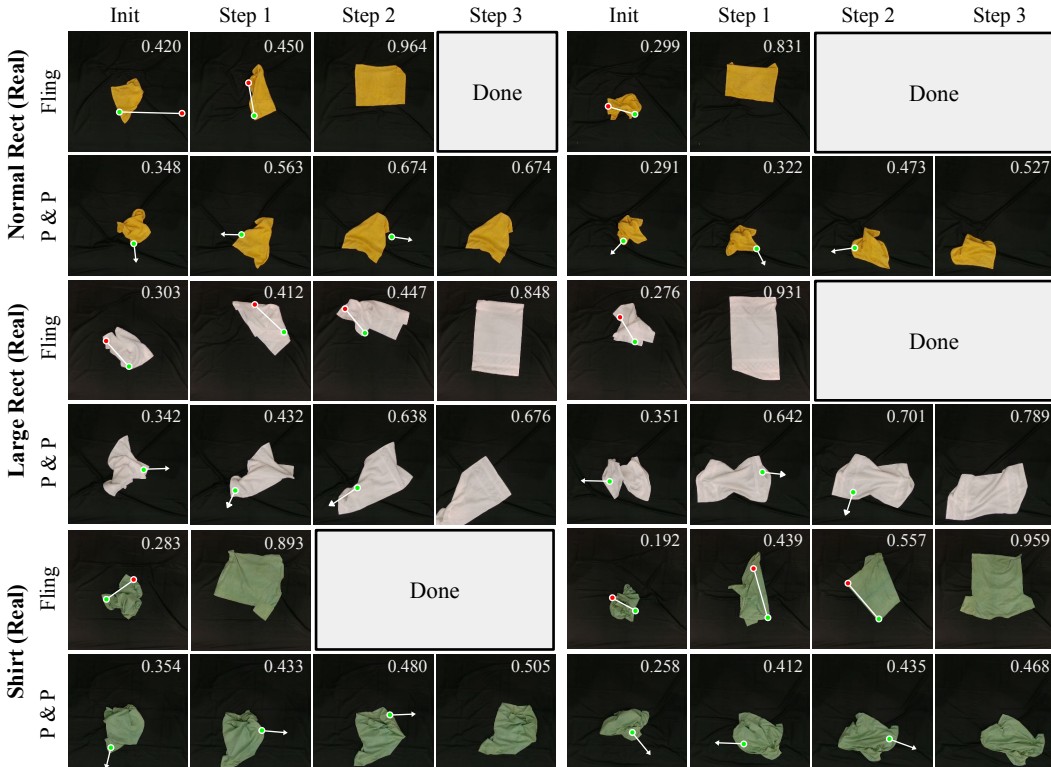

Figure 4: **Qualitative Results in Real World.** Cloth coverages are labeled on the top right corner. Red and green circles represent grasps by left and right arms, placed above and below field of view, respectively. FlingBot discovered through trial-and-error a two-arm corner and edge grasp when corners and edges are visible. While the pick & place baseline discovered a similar strategy, it requires significantly more steps to achieve a coverage lower than FlingBot's.

to real. We manually tuned our simulation fling speed to qualitatively match the real-world cloth dynamics (see supplementary materials on real world robustness to fling parameters).

**Real World Setup.** Our real-world experiment setup consists of two UR5s, where one is equipped with a Schunk WSG50 and the other with an OnRobot RG2, facing each other and positioned $1.35$m apart. The top-down RGB-D image is captured with an Intel RealSense D415. To help with pinch grasp success in real, we used a 2-inch rubber mat as the workspace and rubber gripper finger covers. To mitigate the reliance on specialized force sensors, we chose to implement stretched cloth detection using only a second Intel RealSense D415 capturing a frontal view of the workspace. By segmenting the cloth from the front camera RGBD view of the workspace, then checking whether the top edge of the cloth mask is flat as a proxy for the cloth being stretched, we can pull the arms' end effectors apart until the top of the cloth is no longer bent.

## 4 Evaluation

We design a series of experiments to evaluate the advantage of dynamic actions over quasi-static actions in the task of cloth unfolding. The system is evaluated on its efficiency (i.e., reaching high coverages with a small number of actions), generalization to unseen cloth types (i.e., flattening different types of T-shirts when only trained on rectangle cloths), and reach range (i.e., on cloths with dimensions larger than its reach range). Finally, we evaluate the algorithm's performance on the real-world setup. Please visit https://flingbot.cs.columbia.edu for experiment videos.

**Metrics** The performance is measured by the final coverage, delta-coverage from initial coverage, and the number of interactions. All our coverage statistics are normalized by the coverage of the cloth in a flattened configuration and can be computed from a top-down camera. To evaluate our policy, we load a task from the unseen test datasets then run the policy for 10 steps or until the policy predicts grasps on the floor.

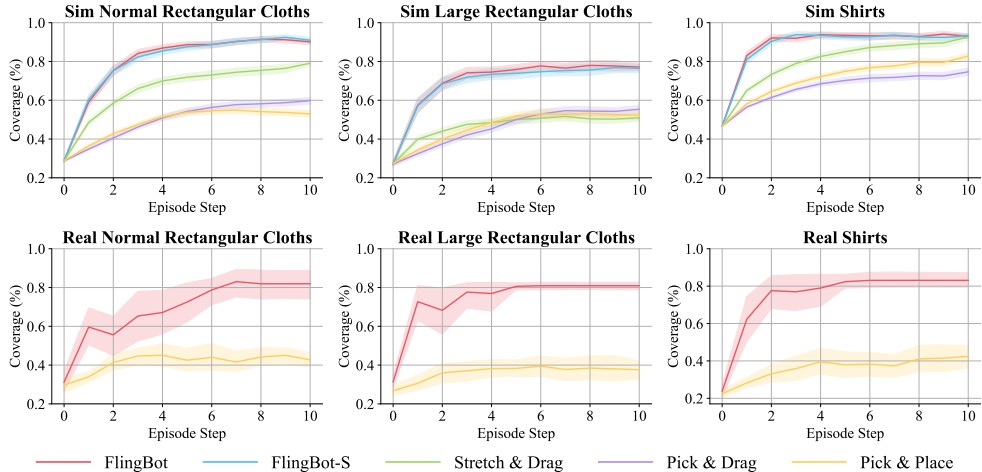

Figure 5: **Coverage v.s. Steps.** With 95% confidence interval shaded. FlingBot can achieve high coverage within a few interaction steps, while the quasi-static baselines never reach high coverages even with significantly more interaction steps. This demonstrates the difficulty of unfolding from highly crumpled initial configurations and dynamic action's superior efficiency.

### 4.1 Task Dataset Generation

Each task is specified by a cloth mesh, mass, stiffness, and initial configuration. The cloth mesh is sampled from one of three types: **(1) Normal Rect**, which contains rectangular cloths with size within the reach range. Edge lengths are sampled from $[0.40\text{m}, 0.65\text{m}]$, **(2) Large Rect**, which contains rectangular cloths with at least one edge larger than the reach range $(0.70\text{m})$, and **(3) Shirt**, which contains a subset of shirts sampled from CLOTH3D's [21] test split, all of which are resized to be within the reach range. The cloth mass is sampled from $[0.2\text{kg}, 2.0\text{kg}]$ and an internal stiffness from $[0.85\text{kg}/\text{s}^2, 0.95\text{kg}/\text{s}^2]$. Finally, the cloth's initial configuration is varied by holding a randomly grasped the cloth at a random height between $[0.5\text{m}, 1.5\text{m}]$ then dropping and allowing the cloth to settle (similar to Lee et al. and tier-3 in Seita et al.), resulting in a severely crumpled configuration. Please refer to supplementary materials for more details on our task generation process.

In simulation, the policy is trained on 2000 *rectangular* cloths sampled evenly between Normal Rect and Large Rect, and evaluated on 600 novel tasks split evenly between Normal Rect, Large Rect, and Shirt cloths. In real, the simulation policy is deployed to collect real world experience on 150 Normal Rect episodes (257 steps), optimized on both simulation and real world data, then evaluated on 10 novel tasks in each cloth type.

### 4.2 Approach Comparisons

We compare against 3 quasi-static baselines, [Pick & Place] (similar to Lee et al. [2], visualized in Fig. 2b), [Pick & Drag] (similar to Seita et al. [22]), and [Stretch & Drag] (identical to [Pick & Drag] with an extra stretch step). Our policy, [FlingBot], predicts the optimal two-arm grasp locations for the flinging primitive (Sec. 3.2). In addition, we compare against [Fling-Reg], which does not exploit the task's equivariances (Sec 3.4) and thus completely fails to perform the task (Tab. 1). Further, we trained [FlingBot-S], which is identical to [FlingBot] but also learns the fling speed. However, Tab. 1 and Fig. 5 show there aren't significant performance gains for [FlingBot-S] over [FlingBot], so we prefer the simpler [FlingBot] for comparisons with baselines. We provide more information about baselines in the supplementary material.

### 4.3 Results

**Better Efficiency.** In this experiment, we compare the unfolding efficiency between quasi-static and dynamic actions. From the Normal Rect column in Tab. 1, [FlingBot] increases the coverage of the cloth $(+63.1\%)$ by more than two times that of [Pick&Place] and [Pick&Drag] baselines $(+29.2\%,$ and $+24.2\%)$. Additionally, from Fig. 5, [FlingBot] achieves over $80\%$ within 3 interactions (simulation normal cloth), while the quasi-static baselines never reach such a high coverage even with

significantly more interaction steps or with a stretching subroutine. This demonstrates the difficulty of unfolding from highly crumpled initial configurations and dynamic action's superior efficiency.

**Increased Reach Range.** In this experiment, we investigate these approaches' performance on Large Rect cloths. These cloths are not only challenging for quasi-static baselines, because the arms can't manipulate the cloth at locations beyond its reach range, but also for our flinging policies, because the arms can't fully stretch or lift the cloth off the ground. Despite these challenges, [FlingBot] achieves 79.2% (Tab. 1, column Large Rect). Compared to the quasi-static baselines, [FlingBot] increases the coverage by +52.0%, which is roughly twice that of the quasi-static baselines ( +27.1%, +24.8%, +23.1%). These results show how high-velocity actions could effectively expand the physical reach range, allowing the system to be more flexible with extreme cloth sizes.

|  | Normal Rect | Large Rect | Shirt |
|---|---|---|---|
| Pick&Place | 53.0 / 24.2 | 52.0 / 24.8 | 79.8 / 33.4 |
| Pick&Drag | 58.0 / 29.2 | 54.2 / 27.1 | 72.4 / 26.0 |
| Stretch&Drag | 77.2 / 48.4 | 50.2 / 23.1 | 90.3 / 43.9 |
| Fling-Reg | 29.1 / 0.3 | 27.7 / 0.5 | 54.0 / 7.6 |
| FlingBot-S | **92.6 / 63.8** | 78.9 / 51.7 | **93.5 / 47.1** |
| FlingBot | 91.9 / 63.1 | **79.2 / 52.0** | 93.3 / 46.8 |

Table 1: Simulation Experiments
(Final / Delta Coverage).

|  | Normal Rect | Large Rect | Shirt |
|---|---|---|---|
| Pick&Place | 43.2 / 13.0 | 38.4 / 11.0 | 42.7 / 19.9 |
| FlingBot | **81.9 / 55.8** | **88.5 / 54.9** | **89.2 / 65.2** |

Table 2: Real World Experiment
(Final / Delta Coverage).

**Generalize to Unseen Cloth Types.** This experiment investigate how well these approaches, trained on only rectangular cloths, can generalize to unseen cloth types (i.e.: T-shirts). Qualitative real world (Fig. 4) and simulation (in supp.) results suggest that our flinging policy has learned to grasp key-points on cloth (i.e., corners, edges) when it sees them, or otherwise fling to reveal these features. FlingBot's generalization performance (93.3% in Tab. 1, column Shirt) to novel cloth geometries can be attributed to this strategy, since a cloth of any type can be unfolded by grabbing one of its edges, stretching, then flinging it in a direction perpendicular to the edge. Through self-supervised exploration, our approach discovered grasping strategies which were manually designed in heuristic based prior works, while being more generalizable to different cloth configurations and types. Meanwhile, all quasi-static baselines exhibited worse cloth unfolding efficiency. The Sim Shirts plot in Fig. 5 shows that our flinging policies take only 3 actions to reach their maximum coverages, while the quasi-static baselines take upwards of 8 steps to reach lower maximum coverages.

**Evaluating Real-World Unfolding.** Finally, we finetune and evaluate our simulation models from Tab. 1 with real-world experience on a pair of UR5 arms. Task generation is automated using the robot arms by randomly grasping the cloth at height 0.50m then dropping it back on the workspace. We use a 0.35m × 0.45m cloth for Normal Rect, a 0.40m × 0.70m bath towel for Large Rect, and a 0.45m × 0.55m T-shirt for Shirt. The system collected 257 experience steps over 150 cloth tasks for finetuning in total. The performance is reported averaged over 10 test episodes, where real-world grasp errors are filtered out (see supplementary materials). From Tab. 2, we report that our policy achieves over 80% on all cloth types, which outperforms the quasi-static pick & place baseline by over 40%. Additionally, we report that the pre-finetune performance of FlingBot on Normal Cloths is 69.8%, justifying our decision to finetune to get a 12.1% improvement. Overall, our flinging primitive takes a median time of 16.7s. While the pick & place primitive only takes a median time of 8.8s, it is unable to reach the high coverages even with many more interaction steps (Fig. 5, bottom row). Both primitives incur additional overheads of 2.5s for preparing the transformed observation batch and 2.9s for a routine which checks whether the cloth is stuck to the gripper after the gripper is opened (details in supplementary materials).

## 5   Conclusion and Future Work

We proposed a fling primitive and a self-supervised learning algorithm for learning the grasp parameters for the cloth unfolding task, resuling in a policy which is efficient, generalizable, and works with cloth sizes beyond the reach range of the system in both simulation and the real world. Future work could explore dynamic actions for more complex (i.e., goal-condition) cloth manipulation tasks by combining with quasi-static action spaces and leveraging cloth pose estimation [23].

**Acknowledgments**

The authors would like to thank Eric Cousineau, Benjamin Burchfiel Naveen Kuppuswamy, other researchers in Toyota Research Institute, Zhenjia Xu, and Cheng Chi for their helpful feedback and fruitful discussions. We would also like to thank Google for their donation of UR5 robot hardware. This work was supported in part by the Amazon Research Award and the National Science Foundation under CMMI-2037101.

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
