# OpenReview forum: "FlingBot: The Unreasonable Effectiveness of Dynamic Manipulation for Cloth Unfolding"
_robot-learning.org/CoRL/2021/Conference — CoRL2021 Oral_

### Official Review · Reviewer_BAzn · 2021-07-18

**Originality:** Very Good
**Technical Quality:** Good
**Clarity Of Presentation:** Good
**Impact:** 3

**Recommendation:**

Strong Accept: I recommend accepting the paper and will argue for my recommendation even if other reviewers hold a different opinion.

**Summary:**

This paper proposes to use dynamic flinging actions with a dual-arm robot for the task of cloth unfolding instead of quasi-static actions. It uses a self-supervised framework based on spatial action maps with the novel parameterization of a fling action primitive. The system takes an image of the scene as input and outputs a value map that provides the parameters for the primitive. The experiments demonstrate that this primitive has better performance than pick and place actions and can be transferred to a real robot; they also demonstrate generalization to different types of cloth.


**Issues:**

- Ablation of pick-stretch-drag instead of pick-stretch-fling
- Test on more types of shirts or other types of clothes in the real world experiment
- Clarify missing details of the method and experiments


**Reviewer Expertise:**

Excellent: Expert knowledge on the topic of the paper

**Strengths And Weaknesses:**

Strengths:
- The proposed parameterization for the dual-arm picking actions is novel and very reasonable.
- The use of the dynamic action is well-motivated in the introduction.
- The method is evaluated on real robot experiments which demonstrate clear improvement over the baselines.
- The self-supervision pipeline is well-constructed to be able to generalize to unseen articles of clothing and does not rely on specific keypoints on the clothes.
- The supplementary video is very helpful.

Limitations:
- The scope/impact of the main novelty (fling motion) is a bit limited to the specific task of unfolding.
- The paper shows the benefit of the fling action primitive which consists of the stretch motion and a fling motion. How much benefit comes from flinging vs stretching? For example, what if the primitive were pick-stretch-drag instead of pick-stretch-fling? Additional ablations on this primitive would strengthen the argument of “the effectiveness of dynamic actions”.
- To better demonstrate the generalizability as well as the failure cases of the method, it would be great to test on more types of shirts or other types of clothes in the real world experiment.
- There are a number of details that are unclear - see below.

Other comments and questions:
- The paper suggests that fling speed prediction does not provide significant performance gain. There might be several reasons for that: (1) The training method could be improved. What if a self-supervised method is used instead of DDPG? For example, the fling speed can be discretized into N bins and the predicted value map can have N channels. (2) The benefit of the fling speed might be more obvious when tested over a wider variety of clothes. A piece of silk might require a very different fling speed compared to a heavy towel. (3) Maybe the fling speed should be correlated with the mass of the cloth which might be difficult to learn from the image.
- How many different cloths are used for the real-world training? How many real-world actions are performed for this data collection?
- Some other figures from the supplementary video can be added to the supplementary text to allow the reader to better understand the method, e.g. cross-over constraint, grasping parameterization; many key parts of the paper were only understood after I saw the supplementary video.

Minor points:
- Fig. 3 is not mentioned in the main text. It could probably be mentioned in Section 3.4 or at the beginning of Section 3.
- Line 213-215 “ The stretching subroutine in our primitive is implemented by pulling the arms’ end effectors apart until the top of the grasped cloth is no longer bent” - how is this computed and implemented?  Details should be added to the appendix.
- Line 264: It would be great to have a brief description of the difference between “pick and place” and “pick and drag”.
- Lines 122-123: “Quasi-static actions, such as pick & place, rely on strategically chosen pick and place points and friction between the cloth and ground to stretch the cloth out. This means these systems must learn complex grasp point selection policies capable of reasoning about the effects of friction on cloth unfolding.” This is not obvious to me - why is friction reasoning important for policies that use pick and place primitives?  Is there a reference to support this claim?  It doesn’t seem like past work that uses pick and place primitives explicitly reasons about friction.
- Line 151 says that L,R are in R^3 and then line 153-154 says that they are in R^2; this is somewhat confusing. What happened to the extra dimension? Was some assumption made here that isn’t explicitly stated?
- Line 162: “ To constrain L to be on the left of R, we can constrain θ ∈[−90◦,90◦]” - a visualization (such as the one in the supplementary video) would be helpful here; this can be added to the supplementary text.
- Line 180-181: “we crop the image such that the cloth takes up at most two-thirds of the image width and height before further scaling it by the 8 scale factors above.” The scale factors are [1.00, 2.75]; this is confusing: if the image is first scaled to take up 2/3 of an image and is later scaled by 2.75, it will then take up 1.83 of the image, e.g. larger than the image size.
- Figure 5 visualization - what do the green and red points each indicate?
- For the top left image in Figure 5, it looks like the red point is not on the cloth.  Shouldn’t this episode get discarded because only one pick point is on the cloth? (Line 337: “When there is a grasp failure in an episode, we discard the episode.”)
- Line 191: “The policy interacts with the cloth until it reaches 10 timesteps or predicts grasps on the floor.” - don’t you mean on the table?
- It is unclear how Flex and Blender are linked; do you import the cloth configuration from Flex into blender for each step in training?
- “Both primitives incur additional overheads of 2.5s for preparing the transformed observation batch” - this seems very slow for just resizing and rotating images.  Is something else going on here?
- “2.9s for a routine which checks whether the cloth is stuck to the gripper after the gripper is opened.” - how does this work?  Details can be added to the appendix.


**Summary Of Recommendation:**

The paper investigated the use of dynamic flinging motion for the cloth unfolding task based on a self-supervised pipeline with spatial action map. The experiments demonstrated the effectiveness of the proposed method and its applicability to the real robot. However, the main contribution of dynamic action is limited to the specific task. Also, it would be great if more experiments can be provided to support the generalizability and the effectiveness of the dynamic action.

# Post-rebuttal Update
I appreciate the clarifications and the additional experiments, such as stretch & drag, which further demonstrates the effectiveness of the fling motion. I would like to keep my current rating. One minor suggestion is to add a section to the Appendix to describe Figure 3 (instead of just adding the figure without any references to it).

---

> ### Author Response · Authors · 2021-08-29
> **Response to Review BAzn**
>
> We thank reviewer BAzn for their thorough feedback. We are encouraged that the reviewer recognized our action parameterization’s novelty. We are also glad they found the supplementary video helpful. We have incorporated their suggestions into the updated paper and visualized failure cases in the supplementary material.
>
> **Ablation on stretching subroutine with quasi-static stretch & drag baseline**:
> We thank the reviewer for the thoughtful suggestion and have added a Stretch & Drag baseline to the list of baselines (Table 1 & Fig. 4). We observed this Stretch & Drag baseline achieved an average final coverage of X% on large cloths (X% worse than FlingBot), verifying our claim that dynamic actions lead to an **effective increase in reach range**. Additionally, Fig. 4 shows that this baseline took significantly more interaction steps to reach the same coverage as FlingBot for normal rectangular cloths and shirts. This result verifies our claim that dynamic actions enable more **efficient** cloth unfolding.
>
>
> > The scope/impact of the main novelty (fling motion) is a bit limited to the specific task of unfolding.
>
> We want to highlight that cloth unfolding is a crucial first step in **many** cloth manipulation pipelines. Future work could explore a combined quasi-static, dynamic action space to exploit dynamic action's effectiveness when appropriate and quasi-static action's flexibility otherwise. For instance, while quasi-static cloth folding from arbitrarily crumpled initial cloth configurations may take a large number of interactions, we expect that a naive addition of Flingbot for initial unfolding to a quasi-static cloth folding system from unfolded cloth configurations should already have better performance and efficiency.
>
>
> >How many different cloths are used for the real-world training? How many real-world actions are performed for this data collection?
>
> We used one normal rect cloth for real world fine tuning data collection, and the real world fine tuning dataset contained 257 points.
>
> **Fling speed variant training method, task cloth physical parameters, and observation space**:
> We appreciate the reviewer’s suggestions for the fling speed variant training method and have similar hypotheses about its lack of advantage over FlingBot. We chose DDPG for being a state-of-the-art off-policy algorithm for continuous action spaces. While cloth physical properties such as mass and stiffness should inform the optimal fling speed, this information is not available from visual observations and is therefore out-of-scope. Leveraging information from other sensors is an interesting future direction.
>
> > For the top left image in Figure 5, it looks like the red point is not on the cloth. Shouldn’t this episode get discarded because only one pick point is on the cloth?
>
> A grasp failure is when the policy predicts a grasp point on the cloth but the real world system fails to execute the pinch grasp. When the policy only predicts one grasp point on the cloth, the system continues with the fling while holding onto the cloth with only one arm.
>
> > It is unclear how Flex and Blender are linked; do you import the cloth configuration from Flex into blender for each step in training?
>
> Yes, we found that launching a Blender subprocess to render and return the observation was around 1 second, which is negligible compare to the amount of time taken to simulate a fling action/ pick & place/ pick & drag action.
>
> > “Both primitives incur additional overheads of 2.5s for preparing the transformed observation batch” - this seems very slow for just resizing and rotating images
>
> We use 12 rotations and 8 scale factors, resulting in 96 transformed images per input observation (256x256) in total.

---

> > ### Comment · Reviewer_BAzn · 2021-08-29
> > **Reply to author**
> >
> > Thank you for your helpful reply!
> >
> > > We used one normal rect cloth for real world fine tuning data collection, and the real world fine tuning dataset contained 257 points.
> >
> > Thank you for the clarification. I would recommend to add this information to the paper somewhere.

---

> > > ### Author Response · Authors · 2021-08-29
> > > **Rely to Reviewer BAzn**
> > >
> > > > I would recommend to add this information to the paper somewhere.
> > >
> > > Thank you for the suggestion. We have updated the paper with this information (L332).

---

> > > > ### Comment · Reviewer_BAzn · 2021-09-03
> > > > **Post-rebuttal update**
> > > >
> > > > I appreciate the clarifications and the additional experiments, such as stretch & drag, which further demonstrates the effectiveness of the fling motion. I would like to keep my current rating. One minor suggestion is to add a section to the Appendix to describe Figure 3 (instead of just adding the figure without any references to it).

---

> > > > > ### Author Response · Authors · 2021-09-06
> > > > > **Response to Reviewer BAzn**
> > > > >
> > > > > Thank you for the suggestion. Section 3.4. was updated during the rebuttal period to reference the stages of Fig. 3.

---

### Official Review · Reviewer_LQc9 · 2021-07-21

**Originality:** Excellent
**Technical Quality:** Excellent
**Clarity Of Presentation:** Excellent
**Impact:** 4

**Recommendation:**

Strong Accept: I recommend accepting the paper and will argue for my recommendation even if other reviewers hold a different opinion.

**Summary:**

This work demonstrates the impressive effectiveness of dynamic fling motion for cloth unfolding.  The parametrization combined with spatial action maps makes it easy and safe to learn dual-arm grasps. The system is trained in a self-supervised fashion, using delta coverage of the cloth as supervision signal. Experiments in simulation and real environments show its efficiency over the pick&place strategy, increased reach range, and generalization to unseen cloth type.


**Issues:**

Issues as stated in the weakness section:

- For sim2real, the authors mentioned mixing sim & real data. It will be very helpful to illustrate how important it is? Since 150 real data points are not much compared to the data in simulation, does it improve a lot or marginally?
- For introducing the spatial action maps, on Page 4, line 169-172. The long sentence makes it difficult to interpret quickly, especially for scaling. Figure 3c&d makes it clearer. Rephrasing the paragraph or adding references & descriptions to the figure can improve the readability of the method.


**Reviewer Expertise:**

Good: General knowledge of the area

**Strengths And Weaknesses:**

Strengths

- The effectiveness of the dynamic fling motion for cloth unfolding is impressive, novel, and beneficial for downstream cloth manipulation.
- The parametrization combined with spatial action maps shows an easy but effective way for learning dual-arm grasps.
- It is appreciated to evaluate the system in both simulated and real experiments thoroughly.
- The paper is well written and enjoyable to read.

Weaknesses
- For sim2real, the authors mentioned mixing sim & real data. It will be very helpful to illustrate how important it is? Since 150 real data points are not much compared to the data in simulation, does it improve a lot or marginally?
- For introducing the spatial action maps, on Page 4, line 169-172. The long sentence makes it difficult to interpret quickly, especially for scaling. Figure 3c&d makes it clearer. Rephrasing the paragraph or adding references & descriptions to the figure can improve the readability of the method.


**Summary Of Recommendation:**

Unfolding a piece of cloth can remove its self-occlusion, which can benefit downstream cloth manipulation tasks. The paper demonstrates the impressive effectiveness of dynamic fling motion for cloth unfolding. The learning diagram is easy but effective for dual-arm manipulation. The experiments are conducted in both simulated and real environments thoroughly.

---

> ### Author Response · Authors · 2021-08-29
> **Response to Reviewer LQc9**
>
> We thank reviewer LQc9 for their thoughtful feedback. We are glad the reviewer acknowledged our approach’s simplicity, effectiveness, and novelty. We are encouraged that they also found our evaluation in both sim and the paper well written and enjoyable to read. We address reviewer LQc9’s concerns below.
>
> > Since 150 real data points are not much compared to the data in simulation, does [mixing sim & real data] improve a lot or marginally?
>
> The average final coverage of normal cloths in the data collection stage (before the model begins finetuning on real world data) was 69.8%, which means finetuning gave a 12.1% improvement. We have updated the paper to include this information.
>
>
> > For introducing the spatial action maps, on Page 4, line 169-172. The long sentence makes it difficult to interpret quickly, especially for scaling. Figure 3c&d makes it clearer. Rephrasing the paragraph or adding references & descriptions to the figure can improve the readability of the method.
>
> We have made the suggested changes to the writing, thank you for the suggestions.

---

> > ### Comment · Reviewer_LQc9 · 2021-09-03
> > **Reply to author**
> >
> > Thank you for the response! That helps a lot to show the effectiveness of the finetuned real data, and provide baselines for future works! I would suggest "strong accept"!

---

### Official Review · Reviewer_X3S4 · 2021-07-23

**Originality:** Excellent
**Technical Quality:** Excellent
**Clarity Of Presentation:** Good
**Impact:** 4

**Recommendation:**

Weak Accept: I recommend accepting the paper, but will not argue for my recommendation if the majority of other reviewers have a different opinion.

**Summary:**

The paper proposes FlingBot, an approach to cloth unfolding using dynamic flinging motions. The key idea of the approach is to grasp a crumpled object with two hands, lift the object and fling it such that it can be placed on the support surface in an unfolded state. This process is repeated several times until the object cannot be unfolded further.

The paper decomposes the problem as follows: (i) a hand-crafted dynamic motion primitive for flinging motions. (ii) a task-specific representation of a bimanual grasp point for flinging. (iii) A learned value network for predicting effective bimanual grasp points for flinging. (iv) A self-supervised learning curriculum for learning the grasp value network in simulation and fine-tune it in real.

The paper shows that the learned dynamic flinging skill results in a more successful, efficient and generalizable unfolding capability then previous pick & place policies for the same task.


**Issues:**

### Writing

- "Delta coverage" is used throughout the text but only defined in Section 4.1. Define it explicitly when introduced for the first time (see next bullet point)
- L52-L56: This explanation is a bit hard to understand when read for the first time. What is a "value map of the same dimension"? What does it mean for a flinging action to be parametrized by a pixel? What is delta coverage? I would recommend to rewrite this paragraph by explaining the approach's idea in simpler terms, and explicitly define all technical terms.
- L75-77 I find that sentence hard to parse. What is the minimally convincing argument? Do the authors mean that demonstrating superior performance on the cloth unfolding task would be a convincing argument? Please reformulate
- L80 Nitpick: The meaning of this sentence is devoid, as the simplicity of an objective generally bears no relationship to the simplicity of the problem.
- L104 State explicitly in which way the task considered here is more challenging (-> more crumpled initial state)
- L123 I don't follow why the network must learn friction effects, please elaborate
- L169-172 I found this section a bit difficult to follow. I would suggest to make explicit references to Fig 3c-e and guide the reader
- L184 It would be helpful to repeat here that training occurs in sim (I was surprised by "task is sampled")


### Real world failure cases

- State explicitly how many episodes had to be discarded because of grasping failures
- Elaborate on how the real world grasp failures could be mitigated in future work

### Technical questions

- Why is the observation so aggressively scaled down to 64x64? I'm wondering how much "crumpling" remains visible in such a downscaled image
- L208 How do you determine "qualitative match"? What is the reason that the simulator behaves so differently/requires half the fling speed. What would happen if the fling speed would not match? Experimental results on the robustness to fling speed in sim would be interesting (if you have some)

### Value network ablation studies

It would be interesting to understand what the value network learns, and in particular how it generalizes. Is it actually paying attention to the item shape or the crumples? Or a mix? If it would only look at shape, it would probably never stop on the T-Shirt task as it has never seen irregularly shapes items during training. Are there irregularly shaped items that would break the network?

I would propose (maybe for an extended journal version of this paper) to systematically feed different item shapes and crumpling states to the network and study the responses of the value network, e.g. whether there are cases where it would infinitely manipulate unfolded cloth, or where it does not notice crumples.


**Reviewer Expertise:**

Very good: Comprehensive knowledge of the area

**Strengths And Weaknesses:**

## Strengths

The paper is very enjoyable to read, and makes an important contribution to the problem of deformable object manipulation in general, and cloth unfolding in particular. It strikes an excellent balance between leveraging general techniques such as reinforcement learning and sim2real and combining them with task-specific motor primitives and representations to solve the task at hand efficiently.

The proposed approach is rigorously evaluated, both in sim and in real, and sensible ablations and baselines are considered.

The figures are beautifully crafted and enable the reader to immediately grasp the key ideas of the approach (Fig 1-3) and how it performs (Fig 5).



## Weaknesses

The paper writing, in particular Sections 1-3 could be more to the point. Some key concepts such as "delta coverage" are used throughout the text but only defined in Section 4.1. Furthermore, the paper states that the real world pipeline suffers from failure cases, but it is not quantified how often these occur, e.g. how many episodes had to be discarded because of a grasping failure. That raises concerns about the practical applicability of the method, and the paper should add a section on how these issues can be mitigated practically.

Furthermore, it would be interesting to add visual ablation studies to understand when the value network breaks down.

**Summary Of Recommendation:**

The paper makes an important contribution to CoRL, both for the concrete task of cloth unfolding and for deformable object manipulation in general. The paper offers a good mix of careful task-specific engineering that enables solving the cloth unfolding task effectively, and a set of general techniques such as self-supervised learning in simulation. To me, this paper constitutes the most impressive account of both simulated and real-world cloth manipulation to date.

---

> ### Author Response · Authors · 2021-08-29
> **Response to Reviewer X3S4**
>
> We thank reviewer X3S4 for their thorough feedback on the writing. We are encouraged that reviewer X3S4 thought our results are the most impressive account of simulated and real world cloth manipulation and our evaluation approach was rigorous. We have incorporated the reviewer’s suggested changes into the paper and address their clarifying questions below.
>
> ### Failure cases in real world and possible mitigations
>
> Most failure cases in our real world experiment were due to failed grasps (where the policy specified a grasp point on the cloth but the grippers failed to pinch grasp the cloth). Cloth grasping failure is a common problem when working with real world cloth manipulation.
> For instance, Ganapathi et al [1] also observed that “the most frequent failure mode is an unsuccessful grasp of the fabric which is compounded for tasks that require more actions”.
> However, we believe this issue can be mitigated by using specialized gripper hardware (like [1,2]) or incorporating grasp success estimation. We have summarized cloth grasping techniques and added them to the discussion in the updated supplementary material.
>
> [1] Ganapathi, Aditya, et al. "Learning Dense Visual Correspondences in Simulation to Smooth and Fold Real Fabrics."
> [2] Seita, Daniel, et al. "Deep imitation learning of sequential fabric smoothing from an algorithmic supervisor." 2020 IEEE/RSJ International Conference on Intelligent Robots and Systems (IROS). IEEE, 2020.
>
>
> > Why is the observation so aggressively scaled down to 64x64? How much "crumpling" remains visible in such a downscaled image?
>
> We are interested in cloth unfolding from challenging cloth configurations that have small initial coverage, for which no prior work achieves a similar level of generality, efficiency, and performance as FlingBot. Intuitively, doing this requires mostly information about coarse grain cloth shapes as opposed to detailed surface wrinkles, and FlingBot’s performance verifies this intuition. In addition, this resolution allows for a larger visual receptive field for the same model architecture, while making training less computationally intensive.
>
> > Fling parameters design and Sim2Real
>
> In designing our motion primitive, we optimized fling dynamics parameters (waypoints, velocities, acceleration) to maximize coverage assuming a good grasp (e.g.: a dual arm grasp on a normal rectangular cloth in a stretched state).
> We observed that the real world flinging setup system could robustly unfold the cloth using a wide range of fling heights, fling speeds, and fling distance for all cloth types from a good grasp, while the simulated system was highly sensitive to such parameters.
> This sim2real gap was bridged in our work by tuning the simulated fling parameters such that flings from good grasps in simulation would also lead to high coverages like in the real system.
> Crucially, this gap underscores the importance of real world results for cloth manipulation as well as motivates future work on fast and accurate cloth simulation engines.

---

> > ### Comment · Reviewer_X3S4 · 2021-09-01
> > **Reply to authors**
> >
> > Thank you for the helpful clarifications and for taking into account the feedback for updating the paper.
> >
> > I have reviewed the new version of the paper and t addressed the majority of my comments were addressed. My only concern is that the I still could not find a note of the number of episodes that were discarded due to grasp failures in the main paper. While I understand that the focus of this paper lies on introducing dynamic motion primitives for the cloth unfolding tasks, the experimental results should be made fully transparent. I would ask the authors to add this information to the main paper (or point me to the line in the paper in case I have overlooked it).

---

> > > ### Author Response · Authors · 2021-09-02
> > > **Response to Reviewer X3S4**
> > >
> > > Grasping failures, where the policy specified a grasp point on the cloth but the grippers failed to execute a successful pinch grasp, constituted all of our real-world pipeline failure cases.
> > > Our real world system automatically detects grasp failures after the dual arm lifts the cloths up using frontal RGBD view.
> > > The average grasp success rate is 78.0%, 45.0%, and 75.8% for normal rectangular, large rectangular, and shirts respectively.
> > > We discard the episodes with a failed grasp, resulting in 281 invalid episodes (184 normal rect, 71 large rects, and 26 shirt).
> > > We used a bath towel for our large rectangular cloths, which is significantly thicker and stiffer than the tea towel and T-shirt we used for normal rect and shirts.
> > > Therefore, pinch grasps with large cloths failed significantly more.
> > > [Here](https://drive.google.com/file/d/1zc0VaBV4EEvhCoQvMXmNdoTrNnqDvAwh/view?usp=sharing) is an anonymized link to visualize all grasp failures.
> > > We will add this information to the final version of the supplementary material.

---

> > > > ### Comment · Reviewer_X3S4 · 2021-09-03
> > > > **Thanks**
> > > >
> > > > Thank you!

---

### Official Review · Reviewer_fiDj · 2021-07-23

**Originality:** Excellent
**Technical Quality:** Very Good
**Clarity Of Presentation:** Excellent
**Impact:** 4

**Recommendation:**

Strong Accept: I recommend accepting the paper and will argue for my recommendation even if other reviewers hold a different opinion.

**Summary:**

The paper proposes a human like, alternative strategy to unfolding cloth: using 2 robot arms to grab the cloth and then fling it (apply a high velocity oscillation). Flinging policies are trained in simulation. Three different categories of cloth are considered. A flinging action primitive is hand designed and a simple but effective action parameterization is used that can overcome some difficulties of dual arm setups. Top down images are captured and fed to a value network, which predicts spatial action maps. The best action parameters are chosen, and the primitive is executed. The methods are compared in simulation against pick & place and pick & drag baselines and show improved efficiency (fewer steps), performance on large cloths, and generalization to unseen cloths. The methods are also compared in real against a pick & place baseline and show improved performance.

**Issues:**

How were the parameters for the dynamic primitive selected? What was that process like? More insight there would be useful.
Is lifting to 0.30 m sufficiently generalizable to long cloth?
How do you know when the cloth is taut during the stretching stage?
Why is learning a velocity unhelpful? When would you expect it to be helpful?
I was confused by how the width and angle of the picking actions are represented or selected from the action value maps. Please clarify this.
All the videos are at 5x speed. Please provide videos at 1x as well.
Were there any notable challenges in sim2real? Would be great to discuss this.

**Reviewer Expertise:**

Fair: Some knowledge of the area

**Strengths And Weaknesses:**

The paper tackles an increasingly popular problem in deformable object manipulation. The method is unique, intuitive, and simple (no state estimation) and seems to perform significantly better than baselines in terms of efficiency, performance, and generalization on complex cloth initializations. Training is done primarily in simulation with fine tuning in the real world. Results are evaluated in both simulation and the real world. The paper is very well written and the video is extremely clear and well illustrated.

As the authors imply in the limitations section, the dynamic flinging strategy and associated findings may be less transferrable to folding cloth than existing pick & place baselines. There are also a few questions. What were the failure cases were in terms of generalization? Which cloths and t-shirts did it fail on and how/why? Please discuss this more. How does performance compare against baselines when the initial state is only lightly or moderately crumpled? Maybe an optimal strategy for unfolding is an initial flinging followed by pick & place. More questions are listed in Issues below.

**Summary Of Recommendation:**

This is a really nice paper on an interesting problem. It provides an unfolding strategy that is clearly the right choice for highly crumpled initial states and demonstrates its effectiveness over benchmarks in simulation and real world. The visuals, video, and clarity also serve as an example for other research work in the field.

---

> ### Author Response · Authors · 2021-08-29
> **Response to Reviewer fiDj (1/2)**
>
> We thank reviewer fiDj for their thorough feedback and clarifying questions. We are encouraged that they recognize our approach's uniqueness, intuitiveness, and simplicity while achieving strong performance over baselines on challenging cloth unfolding tasks. We’ve added the extra information to the paper and addressed their questions below.
>
> ### Quasi-static v.s. dynamic manipulation for cloth unfolding
>
> > How does performance compare against baselines when the initial state is only lightly or moderately crumpled?
>
> Our Fig. 4 shows that dynamic actions can also handle easy cloth initial configurations (starting from high coverages), thus superseding the unfolding abilities of the quasi-static baselines.
>
> > The dynamic flinging strategy and associated findings may be less transferrable to folding cloth than existing pick & place baselines.
>
> While quasi-static cloth folding from arbitrarily crumpled initial cloth configurations may take a large number of interactions, we expect that a naive addition of Flingbot for an initial unfolding stage to a quasi-static cloth folding system from unfolded cloth configurations should already have better performance and efficiency.
> Future work could explore a combined quasi-static, dynamic action space to exploit dynamic action's effectiveness when appropriate and quasi-static action's flexibility otherwise.
>
>
> ### Fling parameters design and Sim2Real
>
> > Were there any notable challenges in sim2real? Is lifting to 0.30 m sufficiently generalizable to long cloth?
>
> To our surprise, the real-world flinging setup system could robustly unfold the cloth using a wide range of fling heights, fling speeds, and fling distance for all cloth types, if given a good grasp (e.g.: dual-arm corner/edge grasp with cloth stretched).
> In contrast, the simulated system exhibits high sensitivity to small changes in these fling parameters.
> This sim2real gap was bridged in our work by tuning the simulated fling parameters such that flings from good grasps in simulation would also lead to high coverages like in the real system.
> Crucially, this gap underscores the importance of real world results for cloth manipulation as well as motivates future work on fast and accurate cloth simulation engines.
>
> Another large sim2real problem was poor collision handling in simulation.
> There were cloth configurations that the real world system experienced but was not observed in simulation, such as cloth twisting.
> While Nvidia Flex is decent at preventing self penetration, it does so at the cost of unrealistic collision handling.
> Qualitatively, this unrealistic collision handling means cloths untwist themselves when twisted (or are in any other state with high self-collision).
> Another case where this self-unfolding behavior was observed was for shirts due to the two layers of the shirts colliding with each other.
> We hypothesize that poor collision handling is the main reason why performance for the simulated shirt benchmark is higher across the board for all approaches, despite being unseen cloth types.
>
>
> > How were the parameters for the dynamic primitive selected?
>
> To achieve the highest speed at the end effector while respecting the torque limit of each joint, the primitive must move upper joints (i.e.: wrist) more than the lower joints (i.e., base). We also found that adding a blending radius between each target joint configuration gave a much smoother flinging trajectory and cloth swinging, as opposed to a jerky cloth motion without blending radius.
> In designing our motion primitive, we optimized fling dynamics parameters (waypoints, velocities, acceleration) to maximize coverage assuming a dual arm grasp on a normal rectangular cloth in a stretched state. Automatically discovering dynamic motion primitives, such as flinging, and simultaneously learning their parameters is an important and interesting direction for future work.
>
> > Why is learning a velocity unhelpful? When would you expect it to be helpful?
>
> We hypothesize that light and thin cloths, whose air resistive forces to momentum during flinging ratio is significantly higher than the cloths tested, would require a higher fling speed. Therefore, fling speed learning may be helpful when cloths in the task dataset contain a wider variance in density and thickness. We also hypothesize that a successful fling speed prediction approach may require extra information about the cloths’ physical parameters that could not be obtained through visual observation alone, which would be out of scope for this work.

---

> ### Author Response · Authors · 2021-08-29
> **Response to Review fiDj (2/2)**
>
>
> > I was confused by how the width and angle of the picking actions are represented or selected from the action value maps.
>
> The grasp width and grasp angle are represented by the rotation and scaling of the input image observations, respectively.
> For instance, if we want to evaluate the action value of more rotations, we would rotate the input image to those rotations and add the rotated images to the batch to evaluate.
> Even though the grasp width and grasp rotation are constant in pixel space (grasp width is always 8 pixels wide, with rotation of 0 degrees), the grasp width and grasp rotation in the world frame change depending on how the image is transformed.
>
> > What were the failure cases were in terms of generalization? Which cloths and t-shirts did it fail on and how/why?
>
> We have visualized some shirt failure cases in the updated supplementary materials. The self-discovered dual-arm corner and edge grasp for flinging which is effective for rectangular cloths fail on shirts in two main ways. First, if the sleeves of the shirt get stuck in the shirt’s collar, FlingBot will be unable to pull the sleeves out. This failure case motivates future work on combining quasi-static and dynamic actions for cloth manipulation. Second, dual arm grasps where one grasp is on the outer surface and another grasp is on the inner surface of the shirt usually flings to low coverages. While this failure case is expected to the differences between rectangular cloths and shirts (presence of holes, inner/outer surfaces, etc.), FlingBot’s performance on shirts still suggested generalizable cloth manipulation abilities.
>
> > How do you know when the cloth is taut during the stretching stage?
>
> To mitigate the reliance on specialized and expensive force sensors, we chose to implement stretched cloth detection by segmenting the cloth from the front camera RGBD view of the workspace, then checking whether the top edge of the cloth mask is flat, as outlined in L214-215.
> We have added these clarifying details to the updated paper.

---

### Meta-Review · Area_Chair_iNcm · 2021-08-10

**Recommendation:** Accept (Oral)
**Confidence:** 4

**Metareview:**

This paper addressed the problem of cloth unfolding with dynamic flinging actions of a bimanual system. At the initial reviews, all four expert reviewers have had a positive rating with three Strong Accepts and one Weak Accept. The reviewers appreciated the simple yet effective approach in this work to enabling new robot behaviors of deformable object manipulation. Some questions have been brought up by the reviewers, including clarifications of model details and discussions on failure cases. The authors provided satisfactory responses to these questions. At the end of the discussion period, the reviewers retained their original ratings. The AC agreed with the reviewers' positive assessments of this work, in particular, the empirical robot behaviors demonstrated in this work. For this reason, we recommended the acceptance of this paper for an oral presentation.

---

> ### Author Response · Authors · 2021-08-29
> **Response to Area Chair iNcm**
>
> We thank the area chair and reviewers for the thoughtful reviews. We have updated the paper, supplementary material, and supplementary video based on the reviewer's feedback and addressed their questions below. Here is the summary:
> 1. Added a quasi-static stretch & drag baseline which stretches the cloth after grasping in the updated paper (as suggested by reviewer BAzn), which verifies FlingBot’s efficiency and increased effective reach range
> 2. Visualized failure cases for all cloth types in the supplementary material
> 3. Discussed when and how fling speed prediction may help
> 4. Described how our dynamic motion primitive parameters were chosen
> 5. Summarized how cloth grasping success could be improved in the real world system in the updated supplementary materials
> 6. Provided clarification on action parameterization, cloth stretching subroutine implementation, observation resolution choice, the effect of finetuning with real world data, grasp failure definition, Blender - PyBullet integration implementation, and image transformation running time
> 7. Incorporated the reviewers’ writing feedback into the updated paper

---

### Decision · Program_Chairs · 2021-09-13

**Decision:**

Accept (Oral)

**Comment:**

This paper addressed the problem of cloth unfolding with dynamic flinging actions of a bimanual system. At the initial reviews, all four expert reviewers have had a positive rating with three Strong Accepts and one Weak Accept. The reviewers appreciated the simple yet effective approach in this work to enabling new robot behaviors of deformable object manipulation. Some questions have been brought up by the reviewers, including clarifications of model details and discussions on failure cases. The authors provided satisfactory responses to these questions. At the end of the discussion period, the reviewers retained their original ratings. The AC agreed with the reviewers' positive assessments of this work, in particular, the empirical robot behaviors demonstrated in this work. For this reason, we recommended the acceptance of this paper for an oral presentation.